# Descriptive Genomic Analysis and Sequence Genotyping of the Two Papaya Species (Vasconcellea pubescens and Vasconcellea chilensis) Using GBS Tools

**DOI:** 10.3390/plants11162151

**Published:** 2022-08-18

**Authors:** Basilio Carrasco, Bárbara Arévalo, Ricardo Perez-Diaz, Yohaily Rodríguez-Alvarez, Marlene Gebauer, Jonathan E. Maldonado, Rolando García-Gonzáles, Borys Chong-Pérez, José Pico-Mendoza, Lee A. Meisel, Ray Ming, Herman Silva

**Affiliations:** 1Centro de Estudios en Alimentos Procesados (CEAP), Talca 3480094, Chile; 2Departamento de Ciencias Vegetales, Facultad de Agronomía e Ingeniería Forestal, Pontificia Universidad Católica de Chile, Santiago 7820436, Chile; 3Laboratorio de Genómica Funcional y Bioinformática, Facultad de Ciencias Agronómicas, Universidad de Chile, Santiago 8820808, Chile; 4Laboratorio de Multiómica Vegetal y Bioinformática, Departamento de Biología, Facultad de Química y Biología, Universidad de Santiago de Chile, Santiago 9160000, Chile; 5Sociedad de Investigación y Servicios, BioTECNOS Ltda., San Javier 3660000, Chile; 6Facultad de Ingeniería Agronómica, Universidad Técnica de Manabí, Portoviejo 130105, Ecuador; 7Laboratorio de Genética Molecular Vegetal, Instituto de Nutrición y Tecnología de los Alimentos, Universidad de Chile, Santiago 7830490, Chile; 8Department of Plant Biology, University of Illinois at Urbana-Champaign, Urbana, IL 61801, USA

**Keywords:** *Vasconcellea*, SNPs/INDELs, GBS

## Abstract

A genotyping by sequencing (GBS) approach was used to analyze the organization of genetic diversity in V. pubescens and V. chilensis. GBS identified 4675 and 4451 SNPs/INDELs in two papaya species. The cultivated orchards of V. pubescens exhibited scarce genetic diversity and low but significant genetic differentiation. The neutrality test yielded a negative and significant result, suggesting that V. pubescens suffered a selective sweep or a rapid expansion after a bottleneck during domestication. In contrast, V. chilensis exhibited a high level of genetic diversity. The genetic differentiation among the populations was slight, but it was possible to distinguish the two genetic groups. The neutrality test indicated no evidence that natural selection and genetic drift affect the natural population of V. chilensis. Using the Carica papaya genome as a reference, we identified critical SNPs/INDELs associated with putative genes. Most of the identified genes are related to stress responses (salt and nematode) and vegetative and reproductive development. These results will be helpful for future breeding and conservation programs of the Caricaceae family.

## 1. Introduction

The family Caricaceae belongs to Brassicales, including papaya, mustard, and Arabidopsis thaliana, among other species [1]. Caricaceae includes five American genera, Carica, Vasconcellea, Horovitzia, Jarilla, and Jacaratia, and one African genus, Cylicomorpha. It has a total of 35 species, most of them described as diploids (2 n = 2 x = 18), with variable reproductive systems including monoecy, dioecy, and polygamy [2,3,4,5].

Carica papaya L. is the most successfully cultivated family and is considered to be the fourth most important tropical fruit crop in the world [6]. However, the Vasconcellea genus, also known as highland papayas or mountain papayas [7], is also relevant for biodiversity and genetics. This is a valuable genetic resource for Carica papaya genetic improvement because some species have shown resistance to PRSV-P (Papaya ringspot virus-P), cold tolerance, and a higher sugar content than tropical papayas [8].

Vasconcellea pubescens (ex V. cundinamarcensis) is a perennial, herbaceous, and trioecy cross-pollinated species with male, female, and hermaphrodite plants [9]. Its life cycle (Figure 1A) begins after emerging from a brief fall-winter break between June and August, when the average temperature reaches >20 °C. In springtime, the growth of new leaves, flowers, and fruits starts, and the fruits of the previous season also begin to mature. This process is maintained continuously between September and May of each year. Female and hermaphroditic plants can reach 4 m in height (Figure 2B) and present white flowers (Figure 2C,D); their fruits are big (180–200 g), spherical to cylindrical, and their color varies from green in immature fruit to yellow in ripe fruit (Figure 2C–E); they contain around 200 seeds.

V. pubescens is cultivated in several South American countries [6,10]. In Chile, it is grown from 29°54′ S–71°15′ W (La Serena, Chile) to 36°08′ S–72°47′ W (Cofquecura, Chile), always under moderate temperatures (between 10 °C and 25 °C) because it cannot tolerate frost or temperatures over 30 °C. For years, its fruits have been used to produce canned fruit, honey, juice, and candies to satisfy the local market, and, recently, these have been exported to the United States and Europe. The origin of V. pubescens is in the Andes Mountain between Colombia and Peru [11]. The time of its introduction to northern Chile is still not precise, but the first Spanish chronicles indicate that native people had already cultivated them before Spanish conquerors arrived in this region around 1535 [12].

Another Vasconcellea species in Chile is V. chilensis (Planch. ex A. DC.) Solms (“Palo Gordo”, native papaya, or austral papaya), a small, endemic, deciduous, and succulent shrub with female and hermaphrodite plants (andro-gynodioecy).

It grows in semi-arid environmental conditions. Its life cycle starts with the development of its leaves at the beginning of April (autumn), when temperatures reach an average of 10 °C, and it loses them in November when the temperature rises above 20 °C. In springtime (September), the first flowers appear, followed by the fruits, which continue until May of the following year. From November to May, the plant without leaves only maintains flowers and fruits (Figure 1B); the photosynthesis activity is carried out through its herbaceous stems.

Female and hermaphrodite plants present small reddish flowers (Figure 3C). Its fruits are small and of various shapes; their color varies from green to brown (Figure 3D), although it is also possible to find some yellow ones; every fruit contains 6 to 7 seeds. Its current fragmented natural populations are between 29°25′ S–71°17′ W and 30°42′ S –71°22′ W (Figure 3A); thus, it has been classified as a vulnerable species [13]. Due to its capacity to resist abiotic and biotic stresses, it is a relevant resource for genetic improvements in Carica and Vasconcellea species [14,15].

Despite their importance, there have been few genetic studies on Vasconcellea species. The additional analysis of these two species will generate valuable information for future conservation and breeding strategies for the Vasconcellea and Carica genus [6]. Most genetic studies have been carried out on Carica papaya [16,17,18]. Co-dominant markers such as simple sequence repeats (SSRs) (i.e., microsatellites) [19,20,21,22] have been identified and used to analyze genetic diversity [16] and develop genetic linkage maps [19,23,24,25,26]. The SSRs isolated in Carica papaya have shown a low transferability to other Caricaceae [27]; thus, these molecular markers have not been used routinely to evaluate Vasconcellea species. Inter-simple sequence repeats (ISSRs) have been used to analyze the genetic structure of cultivated and natural populations of V. pubescens and V. chilensis, respectively [28,29]. However, the genetic information obtained from dominant markers (for example, ISSR) has limited use in genomic applications because the homology of co-migrating amplification products will need to be clarified, and the frequency of heterozygotes is unknown because they are indistinguishable from homozygotes [16,30,31].

Next-generation sequencing approaches (NGSs) have enabled the release of whole-genome reference sequences for many plant species [32,33]. On the other hand, the draft genome of Carica papaya L. is available as a model for tropical fruit trees [1,34]. In this regard, the genome size of C. papaya is 372 Mbp, where 24,746 genes have been identified, with an average gene length of 2373 bp per gene. Additionally, it has been estimated that repetitive sequences accounted for about 52% of the papaya genome, where transposable elements are the most abundant (51.9%), followed by microsatellite sequences (0.19%) [1,34].

The use of next-generation sequencing technology has revealed hundreds of thousands of single nucleotide polymorphisms (SNPs), which are the most abundant type of polymorphism found in eukaryotic genomes [35,36]. Unlike microsatellites, the power of SNPs comes not from the number of alleles but from a large number of loci assessed simultaneously and their genome-wide distribution.

A genotyping method based on partial genome sequencing has been developed, which uses restriction enzymes to digest the genome, reducing its complexity; after the DNA digestion, barcoded adapters are linked to the DNA fragments, which are then sequenced by high-throughput methods, obtaining hundreds of thousands of SNPs simultaneously [37,38]. Those methods have been called reduced-representation sequencing, one of which is genotyping by sequencing (GBS).

The GBS approach is well suited for genetic analysis because it provides useful information for developing conservation and breeding strategies. This information can help to identify priority populations to be conserved, select proper parental lines, and identify advanced breeding lines from the progenies. Additionally, the published reference genome of C. papaya has enabled the analysis of the physical location of the SNPs and INDELs on the genome, showing the relationships of collinearity and synteny between related species [39,40]. GBS has been used to study several cultivated plant species [41], such as barley [37], wheat [42], rice [43], soybean [44], potato [45], apricot [46], peach [47], Chrysanthemum [48], and Japanese plum [49].

The use of mass sequencing tools and the Carica papaya genome sequence as a reference can be a valuable alternative to identify molecular markers such as SNPs and INDELs in Vasconcellea species. To date, there has been no report describing the SNP/INDEL diversity in any species of the family Caricaceae. Thus, a comparative genomics analysis among related Vasconcellea species would facilitate the identification of biologically relevant polymorphisms to study the evolutionary processes affecting their natural and cultivated populations and enable the development of molecular markers for breeding and conservation programs [50].

This study analyzed the genetic diversity of V. chilensis and V. pubescens using SNPs/INDELs and identified some critical SNPs/INDELs associated with putative genes for future conservation and breeding programs for the Caricaceae family.

## 2. Results

### 2.1. Genetic Diversity and Structure

The C. papaya genome (372 Mb) was used as a reference to identify SNPs/INDELs. In this regard, the GBS had 5 Mb and 4.7 Mb coverage for V. pubescens and V. chilensis, respectively. The sequencing depth was 14X for V. pubescens and 27X for V. chilensis.

From 92 samples of V. pubescens, 232,682,246 reads were obtained by GBS. Of them, 89% contained barcode and restriction sites. Finally, 14,213,301 reads were accepted in TOPM. After quality filtering, 4675 SNPs/INDELs were used for genetic analysis (Appendix A).

Low levels of polymorphic loci (p = 62% [se] = 1.12%) and heterozygosity (0.04) were found in cultivated V. pubescens (Table 1). The average nucleotide diversity per site (π) was 0.0032. Heterozygosity and nucleotide diversity did not exhibit significant differences between populations. Tajima’s D test showed a negative value (D = −2.281) and was significantly different from zero according to the beta distribution at 95% confidence [51]. The F_IS_ value was significant (Fis = 0.50 standard error [se] = 0.006), indicating deviation of the Hardy–Weinberg equilibrium and heterozygous deficits; additionally, the average selfing rate (s) was high (s = 0.67), suggesting a high level of inbreeding.

According to the analysis of molecular variance (AMOVA), most of the SNPs/INDELs were distributed among the studied individuals (97%), and the genetic differentiation was low among orchards (ϕPT = 0.03; *p* < 0.001).

In *V. chilensis*, the number of samples by population was 26, 27, 24, and 17 from “Chungungo”, National Park “Fray Jorge”, “Huachalalume”, and Historical Park “Valle del Encanto”, respectively. It is relevant to highlight that this sample size was small because *V. chilensis* is scarce through its remnant natural populations. In total, 310,093,649 reads were obtained, and, finally, 257,142,135 reads were accepted using TOPM. After filtering, 4451 SNPs/INDELs were obtained for population genetic analysis (Appendix A). In *V. chilensis*, a low level of polymorphic loci (*p* = 63.8% [se] = 3.16%) was estimated, but a moderate level of heterozygosity (0.12) was observed (Table 1). Additionally, the inbreeding coefficient (F_IS_) and selfing rate were 0.18 and 0.31, respectively. The nucleotide diversity per site (π) was 0.0154, and it was similar between populations. Tajima’s D neutrality test showed a negative value (D = −1.065) and was not significantly different from zero according to the beta distribution at 95% confidence.

The SNPs/INDELs were mainly distributed among individuals (98%); only a 2.6% genetic differentiation was observed between the populations (ϕPT = 0.026; *p* < 0.001).

### 2.2. Comparative Genomics

Genomic analysis allowed us to identify 730 common polymorphic SNPs/INDELs between *V. pubescens* and *V. chilensis*. Those SNPs/INDELs were subjected to AMOVA to determine their relevance for the interspecific and intraspecific genetic differentiation (ϕPT). SNPs/INDELs were selected when they reached ϕPT values equal or superior to 0.15 after 1000 permutations (*p* < 0.001). The AMOVA identified 58 SNPs/INDELs (Appendix A), which explained 53% (ϕPT = 0.53 *p* <0.001) of the genetic differences between *V. pubescens* and *V. chilensis*. The Bayesian analysis and Evanno methodology for those 58 SNPs/INDELs indicated that the best estimation of delta K was reached when the K value was equal to 2 (delta K = 430.049 for k = 2), clearly separating both *Vasconcellea* species (Figure 4A).

The *Carica papaya* genome (ASGPBv0.4) was used as a reference to locate the 58 SNPs/INDELs over the exon, intron, and intergenic regions. Most of the SNPs/INDELs (72%) were positioned over exons, 17% were over introns, and only 11% were located in intergenic regions (Appendix A). The heterozygosity (Ho) for those 58 SNPs/INDELs was 26% for *V. pubescens* and 52% for *V. chilensis*, showing a negative correlation (r= −0.316; *p* = 0.0156) between both species. Out of the 58 SNPs/INDELs, 52 were positioned over 30 genes according to the *Carica* genome. Some of those SNPs/INDELs reported extreme genetic differentiation values (ϕPT > 0.5; *p*) between *V. pubescens* and *V. chilensis*: tubulin alpha-2 chain (ϕPT = 0.91); phosphatidyl inositol monophosphate 5 kinase (ϕPT = 0.91); nucleoporin 156 (ϕPT = 0.89); cyclic nucleotide-regulated ion channel family protein (ϕPT = 0.88); remorin family protein (ϕPT = 0.84); autoinhibited Ca^2+^-ATPase 1 and 2 (ϕPT = 0.77); signal transduction histidine kinase; hybrid-type, ethylene sensor (ϕPT = 0.73); S-adenosyl-L-methionine-dependent methyltransferase superfamily protein (ϕPT = 0.70); nucleoporin 155 (ϕPT = 0.67); DNA repair (Rad51) family protein (ϕPT = 0.62); sec7 domain-containing protein (ϕPT = 0.61); Deoxyxylulose-5-phosphate synthase (ϕPT = 0.57); and disease resistance-responsive (dirigent-like protein) family protein (ϕPT = 0.52).

When the orchards of *V. pubescens* were compared, 33.4% (ϕPT = 0.334, *p* < 0.001) of the genetic differentiation was determined by 28 SNPs/INDELs (Appendix A), and the Bayesian analysis identified two clusters (K = 2; delta K = 608.078; Figure 4B). Out of the 28 SNPs/INDELs analyzed, 27 were located in intergenic regions, and only 1 SNP was found over an exon (PAC: 16403913.CDS.1) of the putative gene that codifies for the protein CK203. This SNP showed a high level of genetic differentiation (ϕPT = 0.615). For *V. chilensis*, 26 SNPs/INDELs (Appendix A) explained 42% (ϕPT = 0.42 *p* < 0.001) of the genetic differentiation among four natural populations. The Bayesian clustering distinguished two groups (K = 2, delta K = 132.676; Figure 4C). The first group includes the southern populations “Valle del Encanto” and “Fray Jorge”, and the second cluster includes the northern populations “Chungungo” and “Huachalalume”. Out of the 26 SNPs/INDELs, 24 were located in intergenic regions. Furthermore, only 2 SNPs/INDELs were found over coding regions—one over an intron (PAC: 16403969) and the other one over an exon (PAC:16403933 CDS.1). These two loci showed a high level of genetic differentiation (ϕPT = 0.306 and ϕPT = 0.44, respectively) and were located in the annotated gene RING/U-box superfamily protein.

## 3. Discussion

Currently, no public information is available for Caricaceae species regarding population genetic diversity using polymorphisms identified by GBS. Some research has been carried out using dominant molecular markers such as RAPD [52,53,54], AFLP [55], ISSR [28,29], resistance gene analogs [21], and co-dominant markers such as SSR [15,18,56].

At the DNA level, primary molecular variations are single nucleotide polymorphisms (SNPs), one to several hundred base-pair insertions or deletions (INDELs), and variations in the number of tandem repeats (i.e., simple sequence repeat: SSR). Although the SSRs are considered a relevant DNA variation for genetic analysis [57], the SNPs and INDELs have gained strong acceptance because new next-generation technologies have made them more accessible to researchers. The main drawback of SNPs/INDELs is related to the possibility that they are bi-allelic, making them less variable than other genomic variants such as SSR. However, they are ubiquitous, high in frequency, and widely distributed throughout the genome. Sequencing technologies and bioinformatics tools have enabled the discovery of thousands of SNPs and INDELs for population structure studies of wild and cultivated plant species and serve as a valuable genetic tool to identify candidate polymorphisms related to phenotypic variation [58,59]. For instance, the power of SNPs/INDELs comes from the large number of loci that can be assessed simultaneously instead of the number of alleles per locus [60]. SNPs identified in papaya species started to be used only in the last ten years. Nucleotide variabilities for ITS, atpB-rbcL, and trnL-F spacers in Jacaratia Mexicana have been used to identify cultivars in papaya [58,61]. More recently, SNPs/INDELs have been used to study sexual determinations in species of the Caricaceae family. In this regard, 14,528 nucleotide substitutions and 965 short indels associated with sexual identity have been identified [62]. Additionally, a low nucleotide diversity has been found in the X-linked region of papaya [63]; SNPs in candidate genes associated with sex determination have been described elsewhere [64].

To the best of our knowledge, our study is the first report on the organization of genomic diversity in *V. chilensis* and *V. pubescens* using SNPs/INDELs obtained by GBS. In this regard, it was possible to identify 4451 to 4675 high-quality SNPs/INDELs with a depth of coverage of 14X (5 Mb) and 27X (4.7 Mb). We found low levels of genetic variation in these two *Vasconcellea* species, which contrasts with the estimation of genetic diversity reported for plant species with similar biological characteristics such as autogamy, allogamy, and seed dispersal by gravity or ingestion [65].

The cultivated *V. pubescens* showed a high inbreeding level (Fis = 0.50), and the selfing rate (s = 0.67) suggested assortative mating. The scarce gene diversity for this species had previously been reported using 114 inter-simple sequence repeat loci (ISSR [28]). The heterozygosity was inferior to that reported for other allogamous fruit species such as peach (Ho = 0.28 [66]) and grape (Ho = 0.39 [67]).

The nucleotide diversity (π = 3.2 × 10^−3^) was similar to those reported for apple (π = 3.7 × 10^−3^ [68]), peach (π = 2.1 × 10^−3^ [66]), grape (π = 5.1 × 10^−3^ [67]), Chrysanthemum (π = 5.5 × 10^−3^ [48]), Loblolly pine (π = 3.9 × 10^−3^ [67]), and *Thymus vulgaris* (π = 3.1 × 10^−3^ [68]).

However, the nucleotide diversity in *V. pubescens* was superior to that found for *Carica papaya* (π autosome = 1.7 × 10^−3^ and π sexual chromosome = 0.38 × 10^−3^ [63]).

The domestication process (selection and seed propagation) may have played an essential role in reducing the genetic diversity for *V. pubescens* in Chile [63], and it could explain the high level of genetic similarity between individuals in each orchard. There are no records of introducing new germplasms from its origin (high mountains of Colombia and Ecuador), which could have reduced the availability of new genetic variants.

The highland papaya (*V. pubescens*) has been cultivated in northern Chile since the 16th century [12]. Interestingly, the seedlings used to establish new orchards are produced from selected plants from the same orchard; this process is repeated every 6 to 8 years. If this propagation system had been applied for a hundred years, it might have affected the level and organization of its genetic diversity. Additionally, crossing between relatives into the orchards could be responsible for the high homozygosity currently observed. Moreover, *V. pubescens* is self-compatible and displays both cross-pollination and self-pollination simultaneously [6,69].

The genetic differentiation observed between northern and southern orchards (ϕPT = 0.03, *p* < 0.001) can be explained because seedlings established in the southern orchards represent a small portion of the genetic diversity of the northern orchards, and the selection practices applied by the growers may have increased the genetic differentiation between them. These results are coincident with those previously reported [28]. Tajima’s D statistic, which is used to detect divergence from neutrality, showed significant negative values (D = −2.281), suggesting that the pattern of genetic diversity observed in *V. pubescens* has been affected by a selective sweep and population expansion after a bottleneck during its domestication [51].

In contrast, the natural populations of *V. chilensis* showed higher heterozygosity (Ho = 0.120) and nucleotide diversity (*p* = 15.4 × 10^−3^) than cultivated *V. pubescens* but also a lower inbreeding coefficient (Fis = 0.18) and selfing rate (s = 0.3).

The genetic differentiation among populations was low but significant (ϕPT = 0.026; *p* < 0.001). The Bayesian analysis identified two genetic groups (K = 2), including individuals from the four analyzed populations. The Mantel test (r(AB) = 0,32 *p* = 0.05) indicated that there is a slight relationship between the geographic location of the populations and the pattern of genetic diversity. The Tajima D neutrality test (D = −1.065) was negative but not significant, suggesting that there is no evidence that natural selection and genetic drift can affect genetic diversity.

*V. chilensis* is a gynodioecious species that favors allogamy and heterozygosity. However, high levels of homozygosity and a high selfing rate were determined. The fragmentation and the small population size can partly explain these results. The highly fragmented populations are separated from each other, 45 to 85 km apart. Our results showed a tendency for isolation by distance between southern (“Valle del Encanto” Historical Park and “Fray Jorge” National Park) and northern populations (“Chungungo” and “Huachalalume”). Therefore, the pattern of genetic diversity in *V. chilensis* may reflect the outcome of its natural history; thus, the current fragmentation of its populations may reflect recent events that have not yet affected its genetic structure.

From a conservation point of view, the populations located in the “Valle del Encanto” and “Fray Jorge” are protected because the former are in a sector declared a National Historic Monument, and the latter are within a National Park. However, the “Huachalalume” and “Chungungo” populations are not protected and thus suffer from intense anthropogenic pressure.

Comparative genomics between species and populations allowed for the tagging of some genes annotated in the *C. papaya* genome. Several genes were found to be associated with abiotic and biotic stress responses and organogenesis. At the interspecific level, 58 SNPs/INDEL explained 53% of the genetic differentiation between *V. pubescens* and *V. chilensis*, of which 90% were tagged over exons and introns of 30 genes, most of them involved in vegetative and reproductive processes associated with adaptations to different stress conditions (Appendix A).

The SNPs/INDEL with the highest significant value of genetic differentiation were tagged over genes such as phosphatidylinositol monophosphate 5 kinase gene, which codes for an enzyme localized in the apical plasma membrane and adjacent cytosolic region of pollen tubes. It has been implicated in vesicle trafficking and cytoskeletal rearrangements during the carbohydrate metabolic process, cellular amino acid metabolic process, and pollen tip growth [70]. Other genes tagged by SNPs/INDELS that have been described in detail are Nucleoporin 155, 156 codes for proteins located in the nucleolus, plasma membrane, chloroplast, and nuclear pore complex, where they are implicated in nucleocytoplasmic transporter activity. The amino acid sequences of the Nup155 orthologous are highly conserved in the evolution of eukaryotes [71]. The S-adenosyl-L-methionine-dependent methyltransferase superfamily protein gene codes for a protein with methyltransferase activity. This gene is located in the Golgi apparatus, playing an important role in shoot development, the homogalacturonan biosynthetic process, the response to cytokinin stimuli, and root development [72]. DNA repair (Rad51) family protein genes code for a protein involved in the homologous recombination and repair of DNA. Its expression is restricted to pollen mother cells in anthers and megaspore mother cells in ovules [73,74]. The Sec7-domain-containing protein gene codes for a protein involved in the specification of apical–basal pattern formation. This protein is essential for cell division, expansion, and adhesion, and it is located in the endoplasmic reticulum and Golgi apparatus. It is related to endosomal trafficking, which plays an important role in regulating plant growth and development in optimal and stress conditions [75]. Tubulin alpha-2 chain is part of a gene family that codes for a structural protein constituent of the cytoskeleton (tubulin complex, cytosol, cell wall, plasmatic membrane). Tubulins are dimeric proteins involved in controlling fundamental processes such as cell division, the polarity of growth, cell wall deposition, intracellular trafficking, communications, and the response to salt stress [76,77]. Cyclic nucleotide-regulated ion channel family protein genes are involved in the defense response and calcium ion import; they enable the diffusion of calcium ions through a transmembrane aqueous pore or channel [78]. They have also been identified as putative heat sensors, which might play a key role in regulating the heat stress response [79] and modifying proteins located in the plasma membrane, where they participate in plant–microbe interaction signals [80] and virus trafficking through plasmodesmata and the plasma membrane [81]. They are also involved in the process of dehiscence and aging tissues, where mature branched plasmodesmata are predominant [82]. Autoinhibited Ca^2+^-ATPase 1 and 2 genes are self-association proteins located in the plasma membrane and involved in calcium-transport, ATPase activity, and calmodulin-binding. They have also been associated with the response to nematode wounding-related signals in leaves and roots [83].

Signal transduction histidine kinase, a hybrid-type gene, codes for a protein located in membrane components, where it can sense ethylene and affect the metabolism of ABA and other plant hormones such as auxin, cytokinins, and gibberellic acid. It has been implicated in flavonol synthesis, seed germination, and salt tolerance [84].

The deoxyxylulose-5-phosphate synthase gene codes for a protein located in chloroplasts involved in the chlorophyll biosynthetic process, isopentenyl diphosphate biosynthetic process, mevalonate-independent pathway, and response to the light stimulus and mycorrhiza interaction [85,86]. The disease-resistance-responsive (dirigent-like protein) family protein gene codes for a protein located in the endomembrane system. It is believed to be involved in the lignan biosynthetic process and defense response [87]. Their evolutionary history could explain the genetic differences observed between *V. pubescens* and *V. chilensis.*

Other SNPs/INDELs with a high value of genetic differentiation were located over other additional genes, of which thirteen have been annotated, and their putative function has been described. Ten of these genes have been involved in plant development, and three have been associated with stress conditions (more details in Appendix A).

Most of the genetic differentiation among populations of *V. pubescens* was determined by SNPs/INDELs located in intergenic regions. Only one was found over an exon (PAC:16403913.CDS.1), which showed high genetic differentiation. This exon was located in the putative gene CK203 (42% homology), which responds to stress conditions mediated by abscisic acid [88].

Similarly, for *V. chilensis*, only two variants were found over coding regions (PAC:16403969 and PAC:16403933 CDS.1); both showed a high level of genetic differentiation and were located in the gene RING/U-box superfamily protein, which encodes a novel E3 ubiquitin ligase that acts as a central negative regulator of floral organ size [89].

*V. pubescens* and *V. chilensis* have certainly been subjected to different events during their natural history. Domestication, selection, genetic drift, and inbreeding have affected the genetic structure of both species in different ways. Most of the genetic differentiation among populations has been observed in intergenic regions, probably associated with the control of gene expression, although it is not possible to rule out random events.

In this study, we used GBS to understand the genomic diversity and population structure of the cultivated *V. pubescens* and wild *V. chilensis* to identify genetic polymorphisms (SNPs/INDELs) associated with interpopulation and interspecific differences between these two papaya species. Further research could focus on the isolation of these genes reported here; moreover, screening of their expression patterns will provide valuable information to carry out future conservation and genomics-assisted breeding plans.

## 4. Materials and Methods

### 4.1. Plant Material

Young leaves were collected from 92 and 94 adult plants of *V. pubescens* and *V. chilensis*, respectively. The samples were frozen in liquid nitrogen and stored at −80 °C until DNA extraction.

Three orchards of *V. pubescens* were collected in the Coquimbo Region (Figure 2A). The orchards were Guallarauco Ltd.a. (32°23′31″ S–71°22′29″ W, 210 masl), Sociedad Agricola HC (29°5′639″ S–71°09′12″ W, 180 masl), and AgroSaturno SA (29°59′25″ S–71°01′38″ W, 264 masl). Additionally, one population was collected from Frutos de Lipimavida Ltd.a., located in the Maule Region (34°84′42″ S–72°14′08″ W, 0 masl).

Currently, only four natural populations of *V. chilensis* represent the most significant numbers of individuals for this species. These populations are located in the Coquimbo Region of Chile (Figure 3A) and were sampled to carry out this study. The populations sampled were Chungungo (29°25′19″ S–71°17′17″ W, 20 m above sea level [masl]), Huachalalume (29°59′52″ S–71°09′06″ W, 485 masl), Fray Jorge National Park (30°38′06″ S–71°34′16″ W, 600 masl), and Valle del Encanto National Historical Monument (30°42′12″ S–71°22′22″ W, 200 masl).

### 4.2. DNA Extraction and Genotyping by Sequencing (GBS)

The DNA of young leaves from 92 individuals of *V. pubescens* and 94 individuals of *V. chilensis* was isolated using a modified CTAB approach [29]. A concentration of at least 100 ng/µL of double-stranded DNA was corroborated using a Qubit 3.0 Fluorimeter (Thermo Fisher Scientific, Waltham, Massachusetts, USA).

GBS protocols were carried out in the Institute of Biotechnology, Cornell University, Ithaca, NY, USA (http://www.biotech.cornell.edu/brc/genomics-facility, accessed on 17 June 2022)) [37]. Libraries were developed using the restriction enzyme PstI (CTGCAG) and two different adapters. Sequencing was performed using Illumina HiSeq2000 (Illumina Inc., San Diego, CA, USA). The raw files (FASTQ) were aligned to the papaya genome version ASGPBv0.4 [34] using the Burrows–Wheeler alignment tool, version 0.7.8-r441 (http://bio-bwa.sourceforge.net/, accessed on 22 April 2021, [90]). A ‘master’ TagCounts file was produced, which was aligned to the papaya genome v1.0, and a Tags on Physical Map (TOPM) file was built, containing the best genomic position of each tag. The barcode information in the original FASTQ files is used to tally the number of times each tag in the master tag list is observed in each sample (‘taxon’), and these counts are stored in a TagsByTaxon (TBT file) file. The information recorded in the TOPM and TBT was used to call SNPs and INDEL. The information obtained for the genotypes was stored as a filtered VCF file for further analysis with Tassel 5.2.52 (https://www.maizegenetics.net/tassel/docs/TasselPipelineGBS.pdf, accessed on 22 April 2021, [38,91]).

### 4.3. Population Genetic Analysis

SNPs/INDELs were identified following the nomenclature defined in the papaya genome v1.0 [34]. Heterozygosity and Fis were estimated using Tassel software v5.2.52, Genealex v6.5 (https://biology-assets.anu.edu.au/GenAlEx/Download.html, accessed on 15 May 2022, [92] and Popgene v1.31 (https://sites.ualberta.ca/~fyeh/popgene.html, accessed on 3 May 2021, [93]). The selfing rate (s) was calculated as s = 2 F_IS_/(1 + F_IS_) × 100 [93].

The nucleotide diversity (π) values were estimated as the mean pairwise difference between two sequences for all possible sequence combinations [45]. The expectation of π (identified as q in this text) was explored using Tassel v5.2.52 software [91], DnaSP (http://www.ub.edu/dnasp/, accessed on 3 May 2021, [94]), and Mega X (https://www.megasoftware.net/, accessed on 3 May 2021, [95]). The confidence limit of D was obtained by assuming the beta distribution [96].

Tajima’s D neutrality test (D) was used to explore the effect of selection and genetic drift on nucleotide diversity [51,97]. The value of D [π − q/√var (π − q)] equals zero under a neutral model. Negative values indicate an excess of rare variants, recent population expansion, and purifying selection. In contrast, positive values suggest a lack of rare alleles, balancing selection, and sudden population contraction [97].

The population structure was inferred by a Bayesian model clustering algorithm using Structure v.2.3.3 (https://web.stanford.edu/group/pritchardlab/structure.html, accessed on 3 May 2021, [98]). Five independent runs of the algorithm were used, assuming values of K from 1 to 10. The run parameters were performed with 100,000 Markov chain Monte Carlo (MCMC) repetitions and a burn-in period of 10,000, assuming an admixture population. The optimal K value was estimated using ∆K methodology [99]. The proportion of ancestry in a given cluster was calculated as an assignment rate (Q), where 0.8 was accepted as a correct assignment to a single cluster. The ∆Ks obtained by Structure Harvester (https://taylor0.biology.ucla.edu/structureHarvester/, accessed on 3 May 2021, [100]) and Pophelper (http://www.royfrancis.com/pophelper/articles/index.html, accessed on 3 May 2021, [101]) were used to estimate the rate of change in the log probability of data between successive K values and the variance of log probabilities. Each Q value plot was developed in Structure Plot v2.0 software (http://omicsspeaks.com/strplot2/, accessed on 3 May 2021, [102].

The SNPs/INDELs that enable differentiation between *V. pubescens* and *V. chilensis* and among *V. pubescens* and *V. chilensis* populations were estimated using AMOVA ϕPT. The ϕPT estimates the proportion of the variance among populations relative to the total variance [103]. The criteria used to select those SNPs/INDELs were ϕPT values ≥ 0.15 (values of ϕPT ≥ 0.15 indicate a high level of differentiation), with a significance level inferior to 0.05 (*p*) and missing data per locus inferior to 0.10. The ϕPT and its significance level for each locus were estimated with 1000 permutations through Genealex v6.5.

According to the *Carica papaya* genome, the selected SNPs/INDELs were used to identify its physical positions and putative genes. The location of each SNPs/INDEL within the exon and intron was determined using a Perl script (www.perl.org (accessed on 10 September 2021), determining the gene structure limits from the *Carica papaya* genome GFF annotation, version ASGPBv0.4 [34].

## 5. Conclusions

GBS technology enabled the identification of 4675 and 4451 SNPs/INDELs in two papaya species: *V. pubescens* and *V. chilensis*, respectively. The cultivated *V. pubescens* had a reduced genetic diversity at the genomic level and small differentiation between the cultivated populations, although it was possible to distinguish slight differences in the population from southern Chile (Lipimavida, Chile). The effect of a population bottleneck during domestication in Chile could explain those results. Therefore, it is necessary to introduce new germplasms from their centers of origin and initiate a breeding program to develop varieties that combine the adaptation of local genotypes cultivated for hundreds of years in Chile with genotypes from Colombia and Ecuador. On the other hand, the fragmented and isolated natural populations of *V. chilensis* had a moderate genetic diversity and low differentiation between populations, although it was possible to appreciate a particular trend associated with their geographic distribution. Its adaptations to the semi-desert conditions of northern Chile (less than 70 mm of annual rainfall) and temperatures that fluctuate between 9 °C and 20 °C make this species a unique and valuable genetic resource. Finally, it is urgent to take measures to conserve its natural populations because they are suffering based on its lower number of plants due to human activities.

## Figures and Tables

**Figure 1 plants-11-02151-f001:**
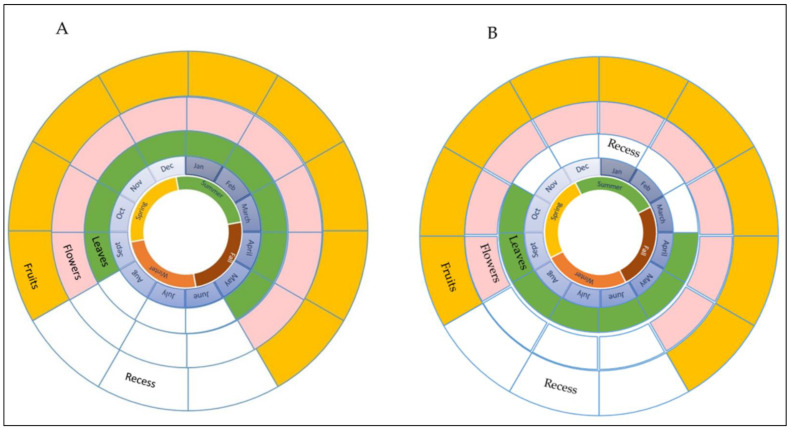
Life cycle of V. pubescens (**A**) and V. chilensis (**B**). Green, pink, and yellow indicate the period of presence and the development of new leaves, flowers, and fruits. White indicates the growth arrest of leaves, flowers, and fruits in V. pubescens, while, for V. chilensis, it indicates the fall of leaves, flowers, and fruits.

**Figure 2 plants-11-02151-f002:**
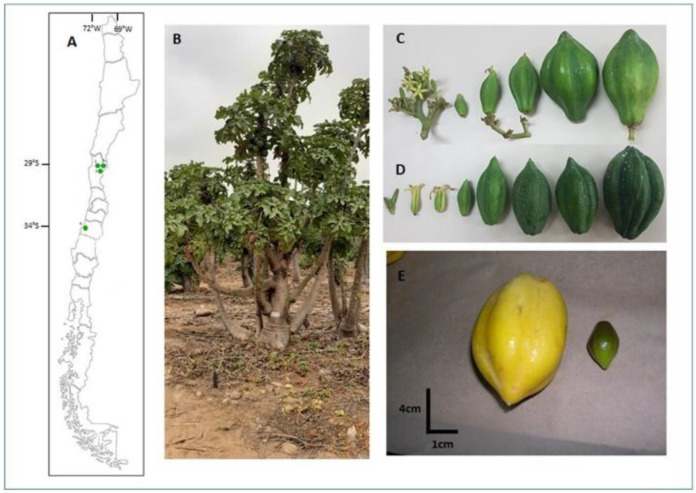
(**A**) Geographic location (29° to 34° South Latitude) of the four orchards of V. pubescens sampled in this study: Guallarauco Ltda, La Serena, Chile (1), Sociedad Agricola HC, La Serena, Chile (2), AgroSaturno SA, La Serena, Chile (3), and Frutos de Lipimavida Ltda, Lipimavida, Chile (4). (**B**) A 4 m-tall plant of V. pubescens grows in an orchard in La Serena, Chile. (**C**) Hermaphrodite and (**D**) female flowers and immature fruits. (**E**) Yellow (V. pubescens, 180 g) and green (V. chilensis, 4 g) ripe fruits.

**Figure 3 plants-11-02151-f003:**
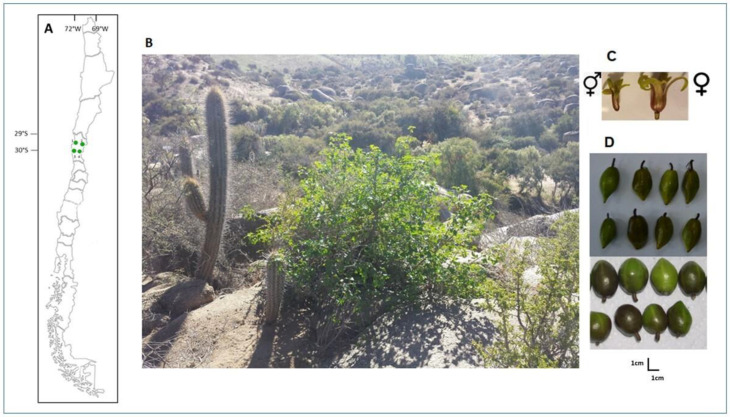
(**A**) Geographic location (29° to 30° South Latitude) of the four natural populations sampled in this study: Chungungo (1), Huachalalume (2), National Park Fray Jorge (3), and Valle del Encanto (4). (**B**) Shrub of V. chilensis grows in its natural semi-desert habitat; leaves can be seen from April to October. (**C**) Female and hermaphrodite flowers with reddish coloration. (**D**) Piriform and oval fruits (3–5 g).

**Figure 4 plants-11-02151-f004:**
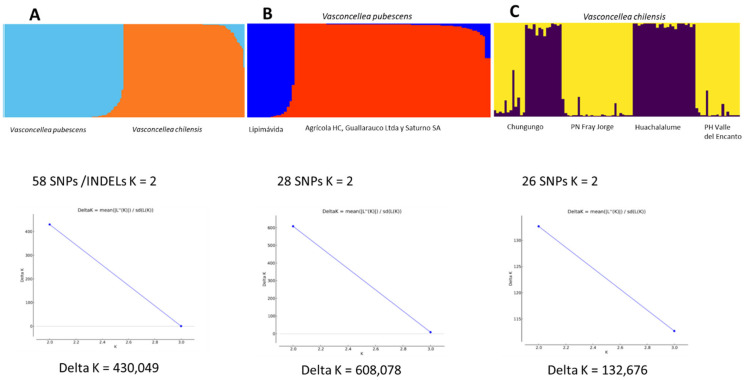
Assignment rate (Q values) using informative SNPs/INDELs. (**A**) *V. pubescens* vs. *V. chilensis* using 58 SNPs/INDELs. (**B**) Cultivated orchards of *V. pubescens* using 28 SNPs/INDELs. (**C**) Natural populations of *V. chilensis* using 26 SNPs/INDELs.

**Table 1 plants-11-02151-t001:** Summary of population genetic statistics for V. pubescens and V. chilensis.

Species	N	L	Ho	π (10^−3^)	D
** *V. pubescens* **	92	4675	0.04	3.2	−2.281
Guallarauco Ltd.a	24		0.05	3.5	
Saturno SA	23		0.04	3.2	
Sociedad Agrícola HC	24		0.04	3.0	
Frutos de Lipimávida	21		0.05	3.1	
** *V. chilensis* **	94	4451	0.12	15.4	−1.065
Chungungo	26		0.12	17.0	
Fray Jorge	27		0.13	17.8	
Huachalalume	24		0.10	15.2	
Valle del Encanto	17		0.12	11.4	

N, number of individuals; L, number of loci (SNPs/INDELs); Ho, observed heterozygosity; π, mean pairwise difference between two sequences; D, Tajima’s D; = π − q/√ var (π − q), where q = expected π.

## Data Availability

Not applicable.

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
