# Peer review of "Descriptive Genomic Analysis and Sequence Genotyping of the Two Papaya Species (Vasconcellea pubescens and Vasconcellea chilensis) Using GBS Tools"

_plants, 2022, doi:10.3390/plants11162151_

Round 1

Reviewer 1 Report

Authors applied GBS technology to compare genetic diversity and discover SNPs between populations of two different species of the Vasconcellea (V. pubescens and V. chilensis), the first one has been subjected to selection for cultivation and the second remained in a wild state. The reference genome for the study is that from Carica papaya.

In my opinion, the manuscript is worthy to be published as it offers genetic resources for papaya breeders. However, I have some concerns about the current form of the manuscript that I will comment to the authors to try to improve it and recommend publication. Importantly, I could not have access to Supplementary material, which could give important information to the manuscript.

            Points to review:

·         Naming of species and genes.

Italics are missing in some cases for abbreviated species name. Please, use always the abbreviated form for C. papaya name (only write Carica papaya the first time).

Genes must be written with capital letters, using the complete name the first time and the abbreviation later on.

·         Figures

Legend of Figure 1 is wrong, please correct “life cicly” and the name of the species, which is repeated.

Legend of Figure 2 is wrong, probably the legend of Figure 3D refers to 3E. I miss some scale bars for pictures 2C and 2D, as well as in Figure 3C and 3D. As a suggestion, I would try to unify Figure 2 and 3, just using those pictures more informative.

·         Results

2.1 Genetic diversity and structure

Regarding the number of filtered SNPs used for analysis, it would be useful to include not only the percentage of SNPs used for analysis and discarded, but the percentage of monomorphic SNPs, or if there are SNPs for which one homozygotic genotype is missing. Describing the kind of SNPs arisen enriched your results.

In the statistical parameters for population genetic description given by the authors the nucleotide diversity per site, the neutrality test and the level of heterozygosity are given. However, I wonder why the authors do not distinguish between expected and observed heterozygosity, and I miss some parameters like allelic richness, which informs about the possibility of finding the two possible alleles in the mayor number of loci.

I have to ask why the authors did not show pairwise genetic differentiation, may be is there a table in the supplementary?

2.2 Comparative genomics

It is very interesting the distribution of SNPs over the different genic and intergenic regions, and could be enticing to find a Figure related in the body of the manuscript, instead of in the supplementary.

·         Discussion

Why is the Mantel test not shown in the manuscript nor in the supplementary?

Lines 420-464. In my opinion, this part of the manuscript needs to be rephrased and shortened. In main cases I just find general descriptions of genes, without a line of discussion. I would use paragraphs for groups of genes.

Did you find some SNPs/INDELS along Transposable Elements?

Materials and methods

4.3. Population genetic analysis

I would include an explanation of the parameter “PT” for genetic differentiation.

Conclussion

Although introgression of characters is mentioned in the Conclussion section, none interesting character is cited.

Author Response

RESPONSE: IT IS IMPORTANT TO NOTE THAT THE WRITTEN ENGLISH WAS REVIEWED THROUGH A SERVICE CONTRACTED TO MDPI. WE CAN ATTACH  THE INVOICE FOR THIS SERVICE IF YOU WANT..

Response to Reviewer 1 Comments

In my opinion, the manuscript is worthy to be published as it offers genetic resources for papaya breeders. However, I have some concerns about the current form of the manuscript that I will comment to the RESPONSE to try to improve it and recommend publication. Importantly, I could not have access to Supplementary material, which could give important information to the manuscript.

RESPONSE: WE ACCESSED THE SUPPLEMENTARY MATERIAL ACCORDING TO THE INSTRUCTIONS PROVIDED BY PLANTS. WE WILL MAKE SURE THAT IT IS AVAILABLE FOR THE READERS.

Point 1: Italics are missing in some cases for abbreviated species name. Please, use always the abbreviated form for C. papaya name (only write Carica papaya the first time).

RESPONSE: Italics were CORRECTED

Point 2: Genes must be written with capital letters, using the complete name the first time and the abbreviation later on.

RESPONSE: WE DO NOT KNOW YOUR SUGGESTION. PLEASE CLARIFY

Point 3: Figures

Legend of Figure 1 is wrong, please correct “life cicly” and the name of the species, which is repeated.

RESPONSE: IT WAS CORRECTED TO “LIFE CYCLE”

Legend of Figure 2 is wrong

RESPONSE:  IT WAS CORRECTED

Probably the legend of Figure 3D refers to 3E.

RESPONSE: IT WAS CORRECTED

Point 4: I miss some scale bars for pictures 2C and 2D, as well as in Figure 3C and 3D.

RESPONSE: IT WAS CORRECTED. A SCALE WAS INCLUDED IN THE FIGURE 2 AND 3

Point 5: As a suggestion, I would try to unify Figure 2 and 3, just using those pictures more informative.

RESPONSE: WE BELIEVE THAT BY UNIFYING FIGURES 2 AND 3, RELEVANT INFORMATION ABOUT BOTH SPECIES WILL BE LOST CONCERNING THE CHARACTERISTICS OF THE PLANTS, FLOWERS, AND FRUITS.

Point 6: Genetic diversity and structure

Regarding the number of filtered SNPs used for analysis, it would be useful to include not only the percentage of SNPs used for analysis and discarded, but the percentage of monomorphic SNPs, or if there are SNPs for which one homozygotic genotype is missing. Describing the kind of SNPs arisen enriched your results.

RESPONSE: IN THIS STUDY, ALL SNPS/INDELS WERE USED REGARDLESS OF WHETHER THEY WERE POLYMORPHIC OR MONOMORPHIC. BOTH SPECIES SHOWED A HIGH LEVEL OF HOMOZYGOSITY, WHICH IS AN INDICATOR OF A LOW LEVEL OF POLYMORPHISM. A DESCRIPTION OF EVERY SNPS CAN BE FOUND IN THE SUPPLEMENTARY MATERIAL 1.

Point 7: In the statistical parameters for population genetic description given by the RESPONSE the nucleotide diversity per site, the neutrality test and the level of heterozygosity are given.

However, I wonder why the RESPONSE do not distinguish between expected and observed heterozygosity, and I miss some parameters like allelic richness, which informs about the possibility of finding the two possible alleles in the mayor number of loci.

RESPONSE: EXPECTED HETEROZYGOSITY AND ALLELIC RICHNESS ARE PARAMETERS ESTIMATED FOR MOLECULAR MARKERS SUCH AS SSR AND DOMINANT MOLECULAR MARKERS. IN THE CASE OF SNPS/INDELS, THE MOST USED AND INFORMATIVE PARAMETERS ARE THE NUCLEOTIDE DIVERSITY (Π), THE EXPECTATION OF Π (IDENTIFIED AS Q IN THIS TEXT) AND TAJIMA'S D NEUTRALITY TEST (D) TO EXPLORE THE DEPARTURE OF NEUTRALITY. THIS INFORMATION IS INCLUDED IN THE TEXT.

Point 8: I have to ask why the RESPONSE did not show pairwise genetic differentiation, may be is there a table in the supplementary?

RESPONSE: WE USED fPT AMOVA TO ESTIMATE THE GENETIC DIFFERENTIATION BETWEEN POPULATIONS. IN THE TEXT (PG 8 AND 9) IS INDICATED THAT FOR V. PUBESCENS AND V. CHILENSIS THE DIFFERENCES BETWEEN POPULATIONS WERE SCARCE (fPT VP= 0.03 AND fPT VCH=0.026).

Point 9: It is very interesting the distribution of SNPs over the different genic and intergenic regions, and could be enticing to find a Figure related in the body of the manuscript, instead of in the supplementary.

RESPONSE: TO AVOID OVERLOADING THE ARTICLE, WE LEAVE THIS INFORMATION IN THE COMPLEMENTARY MATERIAL.

Point 10: Why is the Mantel test not shown in the manuscript nor in the supplementary?

RESPONSE: IT WAS INCLUDED IN THE TEXT

Point 11: Lines 420-464. In my opinion, this part of the manuscript needs to be rephrased and shortened. In main cases I just find general descriptions of genes, without a line of discussion. I would use paragraphs for groups of genes.

RESPONSE: This part was rephrased and shortened.

Point 12: Did you find some SNPs/INDELS along Transposable Elements?

RESPONSE: Unfortunately, we did not find SNPs associated with TE.

Point 13: I would include an explanation of the parameter “PT” for genetic differentiation.

RESPONSE: IT WAS CORRECTED IN THE TEXT PP 14. THE PARAMETER IS fPT. IT IS ESTIMATED WITH AMOVA

Point 14: Although introgression of characters is mentioned in the Conclusion section, none interesting character is cited.

RESPONSE: IN THE SECTION CONCLUSIONS, WE MENTIONED IT IS NECESSARY TO INCREASE THE GENETIC DIVERSITY OF V, PUBESCENS FOR THAT, IT IS NECESSARY TO INTRODUCE NEW GERMPLASM FROM THE ORIGIN CENTER (COLOMBIA AND ECUADOR).

Reviewer 2 Report

the manuscript tilted with" Descriptive genomic analysis of two papaya species (Vasconcel-lea pubescens and Vasconcellea chilensis) based on Genotyping by Sequencing (GBS)".  I see that the manuscript is one of the best articles used the GBS in genotyping and genome analysis for plants. But I have some comments such as;

The title should be changed into; Descriptive genomic analysis and sequence genotyping of the two papaya species (Vasconcel-lea pubescens and Vasconcellea chilensis) using GBS tools.

Abstract

Is good and I having no comment except the phrase; Most identified genes are related to stress responses and vegetative and reproductive development. What type of stresses and is there is a high variation in conditions that these plants are grown? If this is the truth they should mention this in the abstract to give a clear image for the reader.

Figure 2

PAGE 5, figure 2

The morphological characters of the two types should be listed in table, such as, the size and the shape of the flower, the size of the fruits in different times along the season, the major contents of each fruit belong specific types, the antioxidant, the phenolic compounds, etc. the morphological characters associated with genotyping will give the identical identification and the major difference between the two types of papaya plants.

 Line 188: What is the E.???? in the figure legend.

Results

Page 8 and 9

The genome analysis based on genotyping based on sequence, I see that the authors should read the transcriptome because they studying populations in different conditions. So, the environmental conditions will show the variation between the examined plants and this will be clear when the RNA was subjected to examination not the DNA. By the studying the RNA they can confirm the phrase; Most identified genes are related to stress responses and vegetative and reproductive development.

Discussion

Is too long and should be summarized

Materials and Methods

DNA of young leaves from 92 individuals of V. pubescens and 94 individuals of V. 512 chilensis was isolated using a modified CTAB approach, Did the authors make genetic pool for all 92 and then make another genetic pool for 94? Or each leaved subjected to DNA extraction in separate manners.

Conclusion

Is written will but it will be better if it shorten a little.  

Author Response

RESPONSE: IT IS IMPORTANT TO NOTE THAT THE WRITTEN ENGLISH WAS REVIEWED THROUGH A SERVICE CONTRACTED TO MDPI. ATTACHED IS THE INVOICE FOR THIS SERVICE.

Response to Reviewer 2 Comments

Point 1: The title should be changed into; Descriptive genomic analysis and sequence genotyping of the two papaya species (Vasconcel-lea pubescens and Vasconcellea chilensis) using GBS tools.

RESPONSE. OK, WE APRECIATE YOU SUGESTION, THE TITLE WILL BE MODIFYED.

Point 2: Is good and I having no comment except the phrase; Most identified genes are related to stress responses and vegetative and reproductive development. What type of stresses and is there is a high variation in conditions that these plants are grown? If this is the truth they should mention this in the abstract to give a clear image for the reader.

RESPONSE: OK, YOU SUGESTION WILL BE INCLUDED IN THE ABSTRACT AND DISCUSSION. THE GENES ARE RELATED TO RESPONSE TO SALT AN NEMATODE STRESS

Point 3: The morphological characters of the two types should be listed in table, such as, the size and the shape of the flower, the size of the fruits in different times along the season, the major contents of each fruit belong specific types, the antioxidant, the phenolic compounds, etc. the morphological characters associated with genotyping will give the identical identification and the major difference between the two types of papaya plants.

RESPONSE: THIS STUDY AIMED TO CHARACTERIZE THE GENETIC DIVERSITY OF BOTH SPECIES USING SNPS/INDELS. AT THIS STAGE, WE DO NOT CONSIDER PHENOTYPIC ANALYSES. IN A NEXT PAPER WE PLAN TO GIVE INFORMATION ABOUT PAPAIN, ANTIOXIDANT ACTIVITY, PHENOLIC COMPOUNDS AMONG OTHERS.

Point 4: Line 188: What is the E.???? in the figure legend.

 RESPONSE: IT WAS CORRECTED IN THE TEXT

Point 5: The genome analysis based on genotyping based on sequence, I see that the RESPONSE should read the transcriptome because they studying populations in different conditions. So, the environmental conditions will show the variation between the examined plants and this will be clear when the RNA was subjected to examination not the DNA. By the studying the RNA they can confirm the phrase; Most identified genes are related to stress responses and vegetative and reproductive development. 

RESPONSE: WE ARE WORKING IN THE RNAseq FOR V.pubencens AND V.chilensis. WE ARE PREPARING A PAPER ABOUT IT.

Point 5: Is too long and should be summarized.

RESPONSE: OK WILL BE SUMMARIZED

Point 6: DNA of young leaves from 92 individuals of V. pubescens and 94 individuals of V. 512 chilensis was isolated using a modified CTAB approach, Did the RESPONSE make genetic pool for all 92 and then make another genetic pool for 94? Or each leaved subjected to DNA extraction in separate manners.

RESPONSE: DNA WAS ISOLATED FOR EVERY SAMPLE. WE OBTAINED DNA FROM 92 V PUBESCENS AND 94 V. CHILENSIS PLANTS.

Point 7: Is written will but it will be better if it shorten a little.  

RESPONSE: WE WILL BE SHORTENING